# Fe–Mn Oxide Composite Activated Peroxydisulfate Processes for Degradation of p-Chloroaniline: The Effectiveness and the Mechanism

**Yu Shi, Panfeng Ma, Lin Qiao and Bingtao Liu \***

School of Environment and Municipal Engineering, North China University of Water Resources and Electric Power, Zhengzhou 450046, China

\* Correspondence: liubingtao@ncwu.edu.cn

**Abstract:** The chemical co-precipitation method was used to prepare magnetically separable Fe–Mn oxide composites, and the degradation of p-chloroaniline (PCA) using $MnFe_2O_4$ activated peroxydisulfate (PDS). The $MnFe_2O_4$ catalyst exhibited highly catalytic activity in the experiments. XRD, FTIR, SEM and TEM were used to characterize the catalytic materials. $MnFe_2O_4$ calcined at 500 °C was more suitable as a catalytic material for PCA degradation. The elevated reaction temperature was beneficial to the degradation of PCA in neutral pH solution. The reaction mechanism of the $MnFe_2O_4$ catalyzed oxidative degradation of PCA by PDS was investigated by free radical quenching experiments and XPS analysis. The results showed that sulfate radicals ($SO_4^{\bullet-}$), hydroxyl radicals ($\bullet OH$) and singlet oxygen ($^1O_2$) may all be participated in the degradation of PCA. XPS spectra showed that the electron gain and loss of $Mn^{2+}$ and $Fe^{3+}$ was the main cause of free radical generation. The possible intermediates in the degradation of PCA were determined by HPLC-MS, and possible degradation pathways for the degradation of PCA by the $MnFe_2O_4$/PDS system were proposed.

**Keywords:** Fe–Mn oxide composites; persulfate; p-chloroaniline degradation; reaction mechanism

## 1. Introduction

In the past decade, the emission of inorganic and organic contaminants from various industries has led to the degradation of ecosystems [1]. Dyes [2], pesticides [3] and antibiotics [4] have received more attention. Among them, p-chloroaniline (PCA) is often used in industry, mainly as an intermediate for the synthesis of azo dyes and chromophenols but also as an intermediate in pharmaceuticals (e.g., chlordiazepoxide, fenadine) and pesticides and as a colorant in the production of color cinema films [5].However, PCA is highly irritating and can enter the human body through the skin, seriously harming the immune, nervous and endocrine systems [6]. Therefore, the direct discharge of wastewater containing PCA can cause environmental pollution, and the study of an effective method for the treatment of PCA-containing wastewater is essential. Ali et al. [3] found the adsorption of fenuron pesticide adsorption on multi-walled carbon nanotubes and their removal in water. Kenawy et al. [2] developed a nano-composite material and tested their adsorption capacity for dyes in aqueous environments. However, adsorption does not realize the mineralization of organic substance [7]. The choice of advanced oxidation technology allows for the simultaneous decomposition and mineralization of organic matter.

Advanced oxidation technology based on sulfate radical has received extensive attention [8,9]. Sulfate radicals ($SO_4^{\bullet-}$) are generated by advanced oxidation technology based on persulfate (PS). Its standard oxidation potential is 2.5–3.1 V. Persulfates include permonosulphate (PMS) and peroxydisulfate (PDS). The standard redox potential is higher than that of $\bullet OH$. The $\bullet OH$ standard oxidation potential is 1.9–2.7 V [10]. The oxidation principle is also similar to $\bullet OH$, which is stable for a longer time. Advanced oxidation is performed

with highly reactive electron-catalyzed PS, which allows for rapid decomposition of organic matter until final mineralization [11].

Among the many activation methods, transition metal activation is of great interest because it is less energy intensive, cheaper and more reusable than other activation methods (e.g., photo-activation, thermal activation, etc.). Transition metal activated PS are generally divided into two main categories, one is the activation of PS by transition metal ions, such as $Ce^{2+}$, $Cu^{2+}$, $Ag^+$, etc., called homogeneous activation. The other is the activation of PS by solid metals, metal oxides (zero-valent iron, etc.), called non-homogeneous activation. Non-homogeneous catalysts rely on their surface coordination of metal ions to activate PS and provide more active sites [12,13]. Li et al. [14] found that CuO/PS system showed good removal effect on ofloxacin and cefadroxil in water with 92% and 80% removal, respectively. Superoxide radical ($O_2^{\bullet-}$) and $SO_4^{\bullet-}$ are the main active substances to remove them.

In the last few years, spinel ferrites (e.g., $CuFe_2O_4$, $CoFe_2O_4$, $MnFe_2O_4$, etc.) have been widely considered by researchers. Compared to other transition metals, iron is widely used because it is cheap and easy to obtain, less polluting to the environment and effectively activates PS [15,16]. In addition, because spinel ferrite is easily separated from the reaction solution, it can be re-used in the catalytic process [17]. Therefore, it is significant to focus on the PS catalyzed by spinel ferrites. [18]. Considering the market price of metals and heavy metal leaching, which causes secondary pollution to the environment, $MnFe_2O_4$ was introduced into the catalytic PS system as a catalyst, and the results showed good catalytic performance. Deng et al. [19] also found that $MnFe_2O_4$ activated PMS could degrade Orange II in water.

Herein, the chemical co-precipitation method was used to prepare magnetically separable Fe–Mn oxide composites. $MnFe_2O_4$ catalyst coupled with PDS can actively degrade PCA over a wide pH and temperature range. Furthermore, the effect of free radical generation on PCA degradation in $MnFe_2O_4$/PDS system was investigated. Finally, based on the identification of the intermediates, the pathway of degrading PCA was explored. The mechanism of PCA degradation by $MnFe_2O_4$ activated PDS was explained.

## 2. Materials and Methods

### 2.1. Reagents and Materials

All chemicals in this work were analytical grade. P-chloroaniline ($C_6H_6ClN$), anhydrous methanol (MeOH), and tertiarybutyl alcohol (TBA) were supplied by Maclean Biochemical Technology Co, Shanghai, China. Potassium persulfate ($K_2S_2O_8$) was purchased from CNW Technology, Germany. N-(1-naphthyl) ethylenediamine hydrochloride, manganese sulphate monohydrate ($MnSO_4 \bullet H_2O$) and ferric chloride hexahydrate ($FeCl_3 \bullet 6H_2O$) were supplied by Comio Chemical Reagent Co, Tianjin, China. Ethylenediamine tetra-acetic acid (EDTA-2Na) was purchased from Solaibao Technology Co., Beijing, China. Sodium azide ($NaN_3$) was provided by Windship Chemical Reagent Technology Co., Tianjin, China.

### 2.2. Preparation of $MnFe_2O_4$

The $MnFe_2O_4$ used in the experiment was prepared by chemical co-precipitation. The preparation process and properties of the catalyst ($MnFe_2O_4$) were carried out based on the previous experimental results [20]. Firstly, $MnSO_4 \bullet H_2O$ and $FeCl_3 \bullet 6H_2O$ were dissolved in deionized water ($Mn^{2+}$:$Fe^{3+}$ = 1:2, molar ratio). Then, NaOH solution was added to make the pH value of the mixed solution reach 11. Finally, it was filtered, and calcined in a muffle furnace.

### 2.3. Degradation Experiments

For catalyst ($MnFe_2O_4$) performance, 20 mg/L PCA was adjusted to the needed pH value with 0.1 mol/L sulfuric acid or sodium hydroxide. After that, 50 mL of 20 mg/L PCA and a certain amount of $MnFe_2O_4$ were added to each of the six conical flasks. A constant temperature shaker (HNY-2102C, honor, Zhengzhou, China) will be used to hold these

conical flasks. After being placed for 30 min to reach adsorption equilibrium, a certain amount of PDS was added and shaken continuously. After 30, 60, 90, 120, 180 and 240 min of reaction, the conical bottles were removed and the concentration of the filtered solution was measured. For the determination of total organic carbon (TOC), the reaction solution was taken at a fixed point in time, filtered and added 1 to 2 drops of sulphuric acid. Then, the treated solution was put into the TOC analyzer (Trace Elemental Instruments, XPERT, Netherlands) for measurement and quenching experiments in which a quantity of quencher was added before a reaction. The experimental steps of HPLC-MS are as follows: at 0 h and 4 h of the reaction, the filtered PCA solution was taken, and extracted three times with methylene chloride. The organic phase was concentrated to near dryness by means of a rotary evaporator and mixed with 5 mL of ultrapure water [19]. Finally, the organic phase was filtered through the organic phase filter head for the determination of HPLC-MS.

*2.4. Analysis Methods*

The pH values were detected by pH meter (PHS-3C, Shanghai Electronic Scientific Instruments Co., Shanghai, China). The concentration of PCA was analyzed by N-(1-naphthyl) ethylenediamine azo spectrophotometric method using UV-VIS spectrometer (N5000, Shanghai Youke Co., Shanghai, China) at a wavelength of 545nm [21]. The HPLC-MS (W2489-QDa, Waters, Milford, MA, USA) with a reversed-phase C-18 column (4.7 × 250 mm) was used to measurement of intermediate products. The acetonitrile/ultrapure water (V/V = 55/45) was used as mobile phase; the column temperature was 40 degrees Celsius; the flow rate was 0.50 mL/min; the fragment ion scanning range was 50–1050 amu; and the mass spectrometer was subjected to electrospray ionization under the 600 V fragmentation voltage. X-ray diffraction (XRD, Dmax 2500V, Bruker Co, Billerica, MA, USA) with Cu K$\alpha$ radiation ($\lambda$ = 0.15406 nm) was used to analyze the crystallinity of the synthesized products between 10° and 80°. The different functional groups of $MnFe_2O_4$ were detected by Fourier-transform infrared (FTIR, Nicolet 380, Thermo Electric Corporation, Boston, MA, USA). X-ray photoelectron spectroscopy (XPS, Thermo Scientific ESCALAB 250XI, Shimadzu Corporation, Kyoto, Japan) was used to analyze chemical composition, element content and valence state of material surface. Scanning electron microscope (SEM, JSM-6700F, JEOL, Tokyo, Japan) and transmission electron microscope (TEM, JEM-200CX, JEOL, Tokyo, Japan) were used to observe the microscopic morphology and structure of catalytic materials.

## 3. Results and Discussion

*3.1. Characterization of $MnFe_2O_4$*

Relevant studies showed that the calcination temperature may affect the structure of the material itself, thus affecting its catalytic activity [22]. As shown in Figure 1a, the X-ray diffraction patterns of $MnFe_2O_4$ catalytic materials at various calcination temperatures. The samples with same composition correspond well to the characteristic diffraction peaks of cubic spinel-type $MnFe_2O_4$ (JCPDS 38–0430, a = b = 8.519 Å, c = 8.54 Å, $\alpha = \beta = \gamma = 90°$) at 200–500 °C, indicating that they have a semicrystal structure. The characteristic peaks at $2\theta$ = 18°, 29.6°, 34.8°, 42.4°, 52.6°, 56.0° and 61.5° were corresponded to (111), (202), (311), (400), (422), (333) and (440) crystal faces of $MnFe_2O_4$, respectively. The same XRD pattern of $MnFe_2O_4$ was reported by Deng et al. [19]. Therefore, the chemical co-precipitation method used in the experiments was effective in preparing pure $MnFe_2O_4$ material. However, the XRD results of the sample at 600 °C showed that other material components may be presented besides $MnFe_2O_4$, because its characteristic peaks can well correspond to $Fe_2O_3$ (JCPDS 33–0664, a = b = 5.0356 Å, c = 13.7489 Å, $\alpha = \beta = 90°$, $\gamma = 120°$) and Mn2O3 (JCPDS 24–0508, a = 9.4161 = Å, b = 9.4237Å, c = 9.4051 Å, $\alpha = \beta = \gamma = 90°$). The diffraction peaks near $2\theta$ = 23.1°, 38.2° and 55.1°, which belonged to the characteristic peaks of (211), (400) and (044) planes of $Mn_2O_3$, respectively. The characteristic peaks near $2\theta$ = 24.1°, 33.1°, 35.6°, 40.8°, 49.4°, 54.0°, 57.5°, 62.4° and 63.9° were corresponded to (012), (104), (110), (113), (024), (116), (018), (214) and (300) planes of $Fe_2O_3$, respectively.

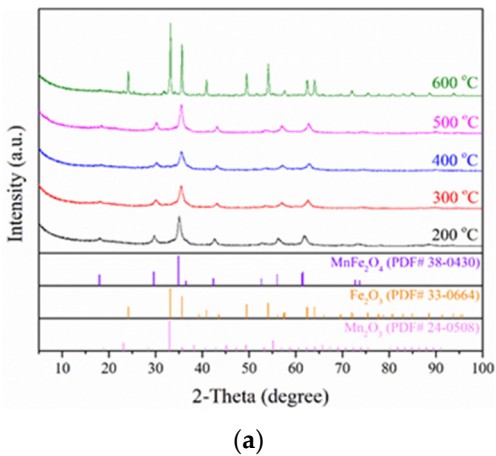
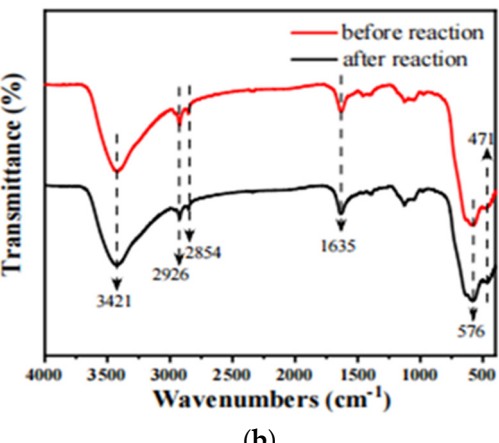

**(a)**                                            **(b)**

**Figure 1.** (**a**) XRD patterns of MnFe$_2$O$_4$ at different calcination temperatures; (**b**) FTIR spectra of MnFe$_2$O$_4$ before and after the reaction.

In order to analyze and determine the changes of surface functional groups of MnFe$_2$O$_4$ before and after the reaction, FTIR spectroscopy was used to analyze MnFe$_2$O$_4$, and the results were presented in Figure 1b. Due to the stretching vibration of hydroxyl and carboxyl groups on the catalyst surface, the larger absorption peak appeared at 3421 cm$^{-1}$. The characteristic peaks also appear at 2926 cm$^{-1}$ and 2854 cm$^{-1}$, which may be related to the C-H extension vibration [23]. Through the bending vibration of water molecules on the catalyst surface, the characteristic peak at 1635 cm$^{-1}$ was generated [24]. In addition, the characteristic peaks at 471 cm$^{-1}$ and 576 cm$^{-1}$, which were resulted in the vibration of Mn-O and Fe-O chemical bonds in the MnFe$_2$O$_4$ catalyst. The low wave number 471 cm$^{-1}$ was related with octahedral coordination of Mn$^{2+}$. The high wave number 576 cm$^{-1}$ was assigned to the tetrahedral coordination of Fe$^{3+}$ [25]. Finally, it was worth noticing that the FTIR of the MnFe$_2$O$_4$ before and after the reaction were basically the same. The prepared MnFe$_2$O$_4$ was a spherical structure with a diameter of 0.1 μm~0.5 μm as derived from TEM images and SEM images (Figure S1).

*3.2. Degradation Experiments in Different Systems*

As exhibited in Figure 2a, it was observed that the effect of adsorption of PCA can be ignored when only MnFe$_2$O$_4$ and PDS were present. In the MnFe$_2$O$_4$/PDS system, the PCA removal rate reached 92% at 240 min. In addition, the degradation efficiency of PCA in the PDS/Fe$^{3+}$ and PDS/Mn$^{2+}$ systems was 5.45% and 13.30% in 240 min, respectively. The results showed that PDS can be activated by MnFe$_2$O$_4$. Similarly, Deng et al. [18] found that when PMS was activated by Fe$^{3+}$ or Mn$^{2+}$ to degrade BPA, the removal rates were less than 10%. It was much more than the 90% removal by heterogeneous activation systems (MnFe$_2$O$_4$/PMS). Figure 2b shows Fe$_2$O$_3$, Mn$_2$O$_3$ and Fe$_3$O$_4$ have lower removal rates than MnFe$_2$O$_4$ under the same conditions. The presence of Mn$_2$O$_3$ and Fe$_2$O$_3$ affect the degradation of PCA. The substitution of spinel tetrahedral positions was confirmed by comparing the MnFe$_2$O$_4$ and Fe$_3$O$_4$ activities.

In this study, the catalytic capability of MnFe$_2$O$_4$ was investigated at various temperatures of calcination and the results were displayed in Figure 3. This calcination temperature of 200 °C to 500 °C had little effect on the formation of its compositions. After 240 min of reaction, the removal of PAC was 90.12%, 85.57%, 87.45%, 92.16% and 80.81% with MnFe$_2$O$_4$ calcined at 200–600 °C, respectively. In addition, the data fitting indicated that the responses followed the quasi-first-order dynamic constants (k$_{obs}$) calculated at 0.012, 0.010, 0.010, 0.013 and 0.009 min$^{-1}$, respectively (Figure S2). These findings showed that the removal rate decreased when the calcination temperature reached 600 °C. The phenomenon may be due to decomposition of the MnFe$_2$O$_4$ material to other oxidizing substances (e.g.,

$Fe_2O_3$ and $Mn_2O_3$) at 600 °C [26,27]. As shown in Figure 2b, $Fe_2O_3$ and $Mn_2O_3$ did not effectively remove PCA. Thus, the degradation rate of PCA was reduced.

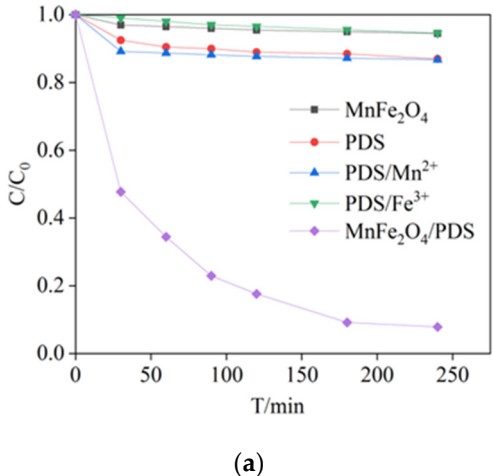
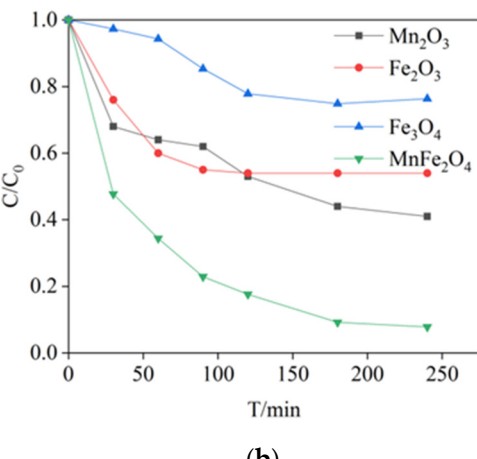

(**a**)                                        (**b**)

**Figure 2.** (**a**) Homogeneous and non-homogeneous conditions for degradation of PCA; (**b**) Degradation of PCA by different solid catalysts. Experimental conditions: [PCA] = 20 mg/L, [$MnFe_2O_4$] = [$Fe_2O_3$] = [$Mn_2O_3$] = [$Fe_3O_4$] = 1.3 g/L, [$Fe^{3+}$] = [$Mn^{2+}$] = 5.6 mM, [PDS] = 2.4 mM and pH = 7.00 ± 0.02.

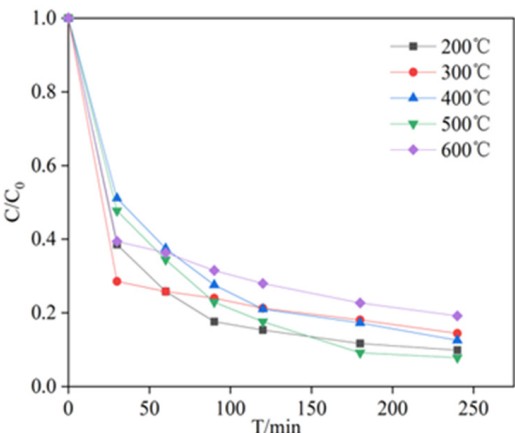

**Figure 3.** Effect of calcination temperatures on degradation of PCA by PDS catalyzed with $MnFe_2O_4$. Experimental conditions: [PCA] = 20 mg/L, pH = 7.00 ± 0.02, [PDS] = 2.4 mM, [$MnFe_2O_4$] = 1.3 g/L, and T = 25 °C.

### 3.3. Effect of Initial pH and Reaction Temperature

The pH of PCA solution has a very significant effect on the degradation rate. Formation of free radicals and surface charge of catalysts were influenced by the pH value [28]. The effect of different pH on the degradation of PCA was shown in Figure 4a. The maximum yield of PCA decomposition was achieved at pH 7, $K_{obs}$ was 0.013 min$^{-1}$ at pH 7 (Figure S3). There were no significant differences in the degradation efficiencies at pH 3, 5, 9. However, when the pH value becomes 11, the efficiency of PCA was reduced by 43.88% compared with that of pH 7.

There are three possible reasons for these results. First of all, as the initial pH of PCA increased from 3 to 11, the leaching concentrations of $Fe^{3+}$ were 0.12, 0.18, 0.14, 0.16 and 0.52 mg/L after 4 h. Meanwhile, the leaching concentrations of $Mn^{2+}$ were 25.82, 19.73, 15.43, 12.25 and 1.71 mg/L. It was inferred that the leaching of $Mn^{2+}$ facilitates the decomposition of PCA, and it also showed that initial pH affects the reaction of $MnFe_2O_4$ and other substances [29]. Secondly, at pH 11, the $SO_4^{\bullet-}$ and $\bullet OH$ were consumed by reaction with $OH^-$ (Equations (1)–(4)), which strongly prevented the degradation of

PCA [30]. For another cause, •OH and $SO_4^{\bullet-}$ have a shorter lifetime in alkaline solutions and therefore cannot adequately capture the bulk phase of PCA [31]. Finally, the charge of PCA and the surface charge of $MnFe_2O_4$ are also a crucial factor influencing the degradation of PCA in solutions. The acidity coefficient value ($pK_a$) of PCA was reported to be 4.15. It meant that PCA was primarily present in solution as a form of cationic at pH 3 and existed as neutral or anion form at pH 5–11 [18]. The point of zer charge ($pH_{pzc}$) of $MnFe_2O_4$ was determined to be 4.71 (Figure S4). It indicated that the catalyst surface is positively charged when $pH < pH_{pzc}$. When $pH > pH_{pzc}$, the catalyst surface is oppositely charged as compared to $pH < pH_{pzc}$ [32]. The pH changes of the solution during reaction were monitored and displayed in Table S1. These findings suggest that the system pH environment was maintained at 4 or below after 30 min of reaction for solutions with initial pH values of 3 to 9. When the $pH_{pzc}$ of $MnFe_2O_4$ is higher than the pH of the solution, it has a positive surface charge. This condition is favorable for the production of more $SO_4^{\bullet-}$. When the pH value increases to 11, the electrostatic gravity effect between $MnFe_2O_4$ ($pH > pH_{pzc}$, negative charges) and PDS (anion) disappears, and the yield of oxidizing active substances decreased, which resulted in a decline in PCA removal yield.

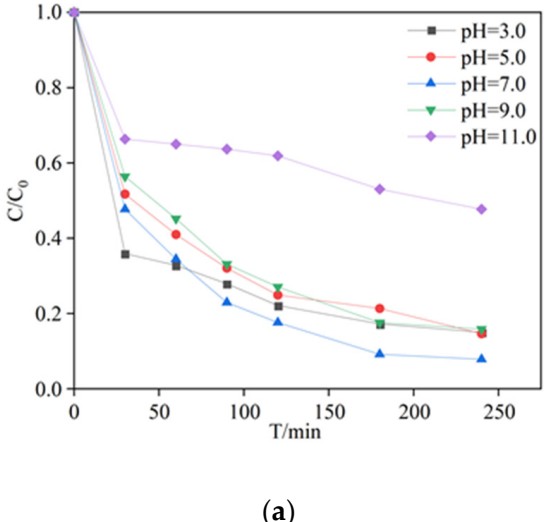 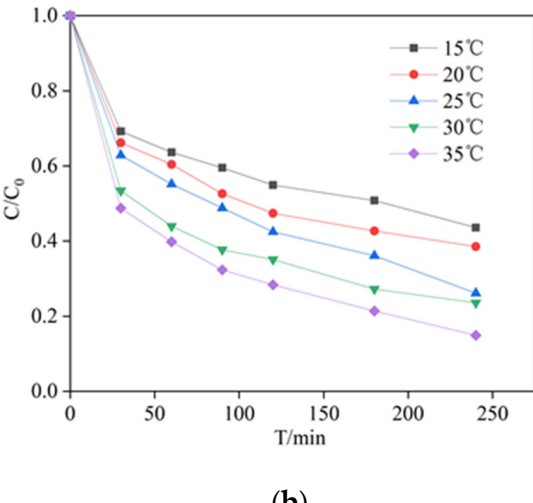

(**a**)                                                          (**b**)

**Figure 4.** The effect of the following factors on PCA concentration ratio: (**a**) initial pH, experimental conditions: [PCA] = 20 mg/L, [PDS] = 2.4 mM, [$MnFe_2O_4$] = 1.3 g/L and T = 25 °C. (**b**) Solution temperature, experimental conditions: [PCA] = 20 mg/L, [PDS] = 2.4mM, [$MnFe_2O_4$] = 0.5 g/L and T = 15–35 °C.

$$SO_4^- + H_2O \rightarrow HSO_4^- + OH \tag{1}$$

$$SO_4^- + OH^- \rightarrow SO_4^{2-} + OH \tag{2}$$

$$OH + OH \rightarrow H_2O_2 \tag{3}$$

$$S_2O_8^{2-} + H_2O_2 \rightarrow 2H^+ + 2SO_4^{2-} + O_2 \tag{4}$$

This study also investigated the degradation of PCA by the $MnFe_2O_4$/PDS system at different reaction temperatures (Figure 4b). As shown, the PCA removal rate gradually increased when the temperature increased. After 240 min of reaction, the removal rate increased by 28.7% with a temperature increase of 20 °C. Therefore, temperature has an important effect in the reaction of PCA removal by catalyst activated PDS. $K_{obs}$ were 0.0041 min$^{-1}$ at 15 °C and 0.0091 min$^{-1}$ at 35 °C, with significantly higher removal efficiency (Figure S5). PCA can be effectively degraded at room temperature. In addition, the Arrhenius equation was used to estimate the relationship between reaction rate and temperature (Equation (5)).

$$lnK_{obs} = lnA - E_a/RT \tag{5}$$

where $E_a$ was the activation energy (kJ/mol), $K_{obs}$ was measured quasi-first-order dynamic constant, R was the universal gas constant (8.314 J/mol·k), A was Arrhenius constant and T was the temperature (K). In accordance with Equation (5), the $E_a$ of PCA removal by $MnFe_2O_4$ was calculated to be 29.70 kJ/mol [20]. Therefore, what occurs in the $MnFe_2O_4$/PDS system is mainly a chemical reaction.

### 3.4. Quenching Experiments of PCA Degradation by MnFe₂O₄/PDS System

The reasons for PCA degradation in the $MnFe_2O_4$/PDS system were investigated by quenching experiments, and the main active species were examined. In quenching experiments, a probe scavenger was used to remove free radicals [33]. Several studies reported that there are two main reactive species (e.g., $\bullet OH$ and $SO_4^{\bullet-}$) (Equations (1), (2) and (6)) [28]. MeOH has a high heat of reaction for both $SO_4^{\bullet-}$ and $\bullet OH$ ($k_{\bullet OH} = 9.7 \times 10^8$ $M^{-1}s^{-1}$, $k_{SO4}^{\bullet-} = 3.2 \times 10^6$ $M^{-1}s^{-1}$). Moreover, the kinetic rate of the reaction of TBA with $\bullet OH$ ($k_{\bullet OH} = (3.8–7.6) \times 10^8$ $M^{-1}s^{-1}$) is faster, compared to $SO_4^{\bullet-}$ ($k_{SO4}^{\bullet-} = (4.0–9.1) \times 10^5$ $M^{-1}s^{-1}$) [34,35].

Figure 5a,b shows the degradation of PCA in the $MnFe_2O_4$/PDS system when TBA, MeOH and PDS were added into solution at the ratios of 200, 500 and 1000. It could be observed that the PCA removal rate decreased significantly after the addition of both TBA and MeOH. In this work, TBA (64.58%) and MeOH (62.18%) had similar effects on the degradation of PCA. Their $K_{obs}$ also decreased from $1.255 \times 10^{-2}$ $min^{-1}$ to $0.506 \times 10^{-2}$ $min^{-1}$ and $0.459 \times 10^{-2}$ $min^{-1}$, respectively (Figures S6 and S7).These findings indicated that both $\bullet OH$ and $SO_4^{\bullet-}$ maybe involved in PCA degradation. In addition to $SO_4^{\bullet-}$ and $\bullet OH$ as the main free radicals, non-radical ($^1O_2$) can also be produced at the same time (Equations (7) and (8)) [36]. As shown in Figure 5c, $NaN_3$ was used to quench $^1O_2$ [37]. $^1O_2$ played the significance role in the system. After adding $NaN_3$, the percentage of degradation decreased from 92.16% to 60.80%. The $K_{obs}$ decreased from $1.255 \times 10^{-2}$ $min^{-1}$ to $0.421 \times 10^{-2}$ $min^{-1}$ (Figure S8).

$$S_2O_8^{2-} + MnFe_2O_4 \rightarrow 2SO_4^{-} \qquad (6)$$

$$S_2O_8^{2-} + 40H^- \rightarrow 2SO_4^{2-} + {}^1O_2 + 2H_2O \qquad (7)$$

$$O_2^{-} + 2H_2O \rightarrow {}^1O_2 + 2H_2O_2 + 2H^+ \qquad (8)$$

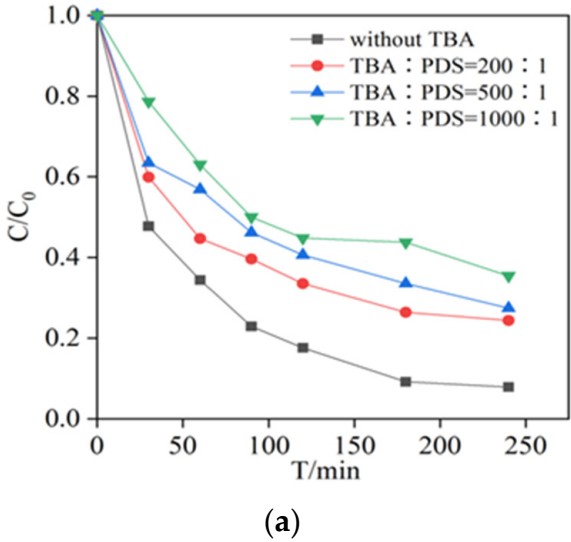

(a)

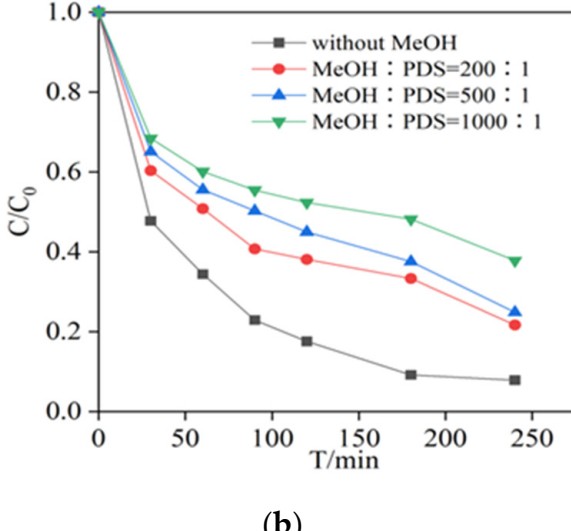

(b)

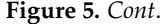

**Figure 5.** *Cont*.

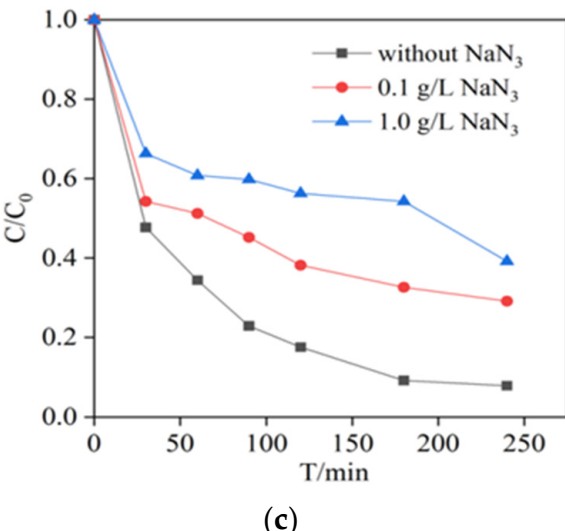

**(c)**

**Figure 5.** Effect of different (**a**) TBA, (**b**) MeOH and (**c**) $NaN_3$ concentration on degradation of PCA by PDS catalyzed with $MnFe_2O_4$. Experimental conditions: [PCA] = 20 mg/L, [PDS] = 2.4 mM, [$MnFe_2O_4$] = 1.3 g/L and T = 25 °C.

### 3.5. Total Organic Carbon (TOC) Removal Efficiencies in MnFe₂O₄/PDS System

A number of studies have shown that it is extremely difficult to obtain the full mineralization of PCA by chemical treatment methods [38]. A high removal rate of organic matter does not mean that the organic matter is broken down into $CO_2$ and $H_2O$, which may also be present as other small molecules [39]. The TOC was measured under different PDS dosages. As displayed in Figure 6, when 0.4 mM of PDS was added into PCA solution, the removal rate of TOC was 23.50% in 240 min. However, the TOC removal rate was 35.07%, when 2.4 mM of PDS was added into PCA solution. The removal and mineralization rates of PCA were significantly increased. The results indicated that PCA could be effectively mineralized in the $MnFe_2O_4$/PDS system. The PCA was not completely mineralized to $H_2O$ and $CO_2$ [40]. It is possible that inorganic or organic small molecule products of oxidation were developed in solution.

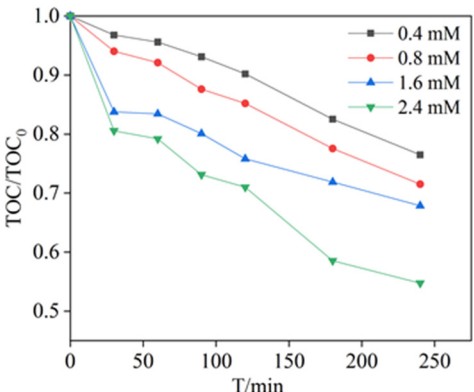

**Figure 6.** Effect of different PDS concentrations on the removal efficiencies of TOC by PDS catalyzed with $MnFe_2O_4$. Experimental conditions: [PCA] = 20 mg/L, [PDS] = 0.4–2.4 mM, [$MnFe_2O_4$] = 1.3 g/L and T = 25 °C.

### 3.6. Mechanism of PCA Degradation by MnFe₂O₄/PDS Systems

To further investigate the degradation mechanism of PCA in the $MnFe_2O_4$/PDS system, the products formed after the oxidative degradation of PCA were identified by HPLC-MS. The HPLC-MS is a concatenated technique using liquid chromatography as the separation system and mass spectrometry as the detection system. HPLC-MS combines

chromatography and mass spectrometry to obtain more quantitative detection results. [41]. It is capable of providing relative molecular mass and structural information for the quantitative analysis of different contaminants [2,42].

Several degradation intermediates of PCA in the $MnFe_2O_4$/PDS system were detected using HPLC-MS measurements. The result obtained was similar with previous studies [43]. The peak area of PCA decreased significantly after 4 h, accompanied by the generation of new peaks, indicating the generation of new intermediates. The mass spectrometry analysis in Figure 7 showed that two major intermediate products may be formed during PCA degradation. The corrected retention time for Peak1 ($P_1$) was 12.77 min and m/z was 158.08. The corrected retention time for Peak2 ($P_2$) was 20.63 min and m/z was 267.08. Compared with PCA, the peak areas of $P_1$ and $P_2$ were reduced by 87% and 28%, respectively. By comparison of the mass to nucleus ratio and its molecular composition, it was assumed that $P_1$ and $P_2$ may be chloronitrobenzene and 5-chloro-2-(4-chloro-phenyldiazene) phenol, respectively.

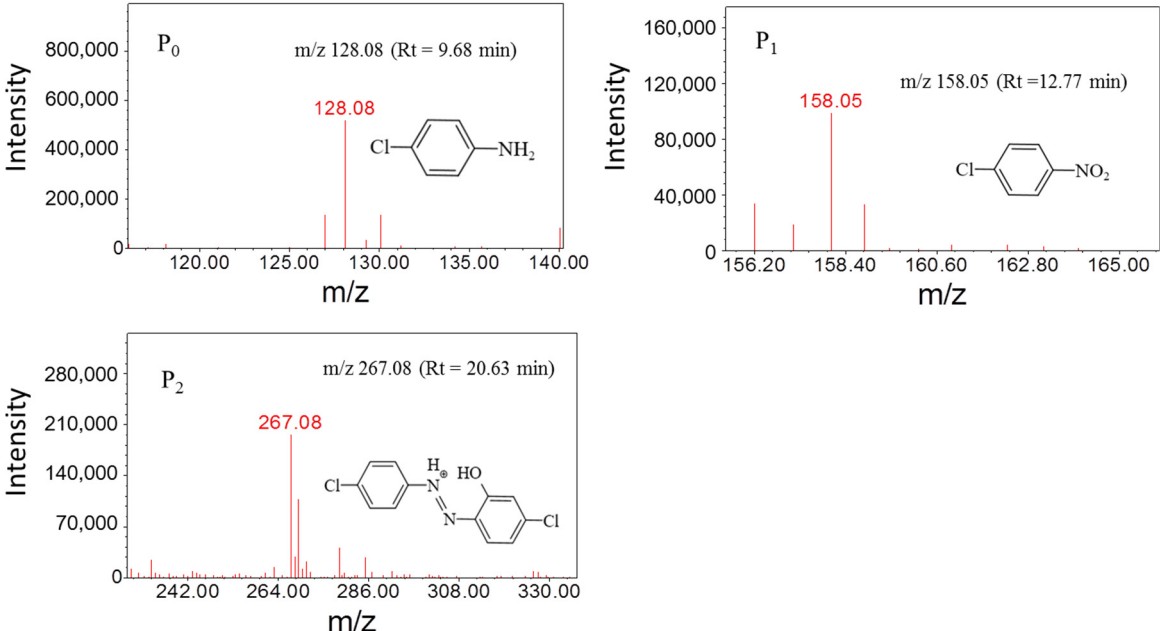

**Figure 7.** Mass spectra of PCA and its degradation products. Experimental conditions: [PCA] = 20 mg/L, [PDS] = 2.4 mM, [$MnFe_2O_4$] = 1.3 g/L.

Combined with previous studies [43–45], two possible degradation paths of PCA in the $MnFe_2O_4$/PDS system were obtained and are shown in Figure 8. In pathway I, deaminization reaction had played an essential part in the process of PCA degradation. This deaminization process would generate B, Then, P1 (chloronitrobenzene) was formed under the attack of •OH. In pathway II, the benzene ring on PCA can form phenolic compounds in the presence of $SO_4^{•-}$, •OH and $^1O_2$, such as 2-amino-5-chlorophenol, which reacts with B to form substance $P_2$ (5-chloro-2-(4-chlorophenyldiazene) phenol). PCA and its intermediate oxidation products ring cleavage reaction occurred in the presence of reactive species and were converted by the oxidation of $SO_4^{•-}$ and •OH to minor organic molecules.

It is well known that although the main structures of the compounds are the same, different substituents have important effects on the properties of the substances. Structural features of the compound determine the solubility, molecular arrangement, space structure, intermolecular attraction and repulsion of organic compound. This affects the degree to which organic matter is catalytically degraded. The compound in this study was PCA. According to the functional group structure analysis of the PCA, when there is already an amino substituent, the benzene ring contains a second substituent, and the second substituent is a halogen group. The halogen group has a passivation on the benzene

ring. Theoretically, this organic is more difficult to degrade than those substances with a single substituent (e.g., phenol, nitrobenzene, chlorobenzene, etc.). Therefore, it is of great significance to study the degradation of different substituent compounds.

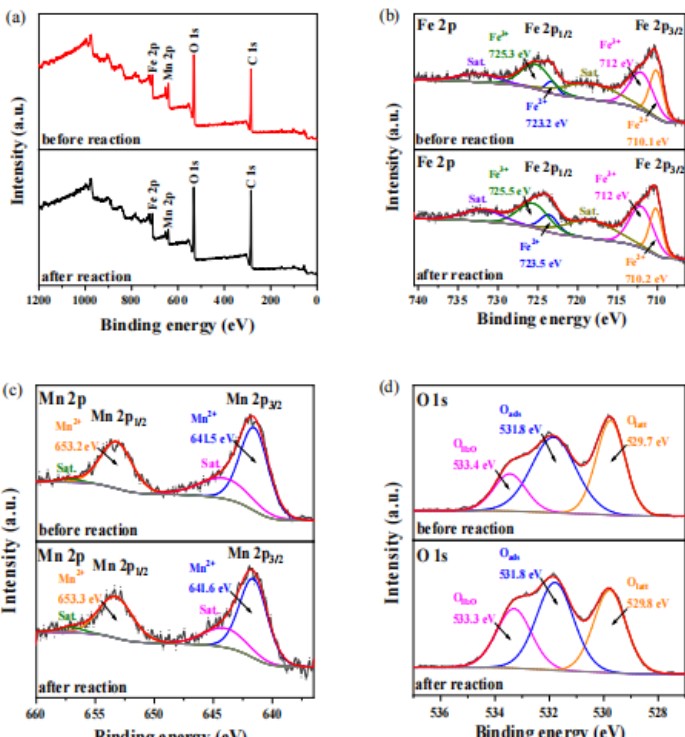

**Figure 8.** Possible degradation pathways of PCA in the $MnFe_2O_4$/PDS system.

This reaction process was a non-homogeneous process. The PCA was adsorbed onto the catalyst surface. The oxidation reaction occurred on the surface of the catalyst. The oxidation during the catalytic activity may lead to changes in the surface composition and chemical state of $MnFe_2O_4$ [46]. Therefore, XPS was used to characterize the surface chemical properties of $MnFe_2O_4$ before and after degradation experiment [47]. The results showed that elements C, O, Fe and Mn exist in $MnFe_2O_4$ (Figure 9a).

**Figure 9.** Full Spectrum of XPS before and after reaction of $MnFe_2O_4$ Catalyst (**a**) and fine XPS spectra of Fe 2p (**b**), Mn 2p (**c**) and O 1s (**d**).

As seen from Figure 9b, fresh sample of Fe 2p, the binding energies (BE) of the $Fe^{2+}$ peaks were 710.1 eV and 723.2 eV, which were accorded with Fe $2p_{3/2}$ and Fe $2p_{1/2}$, respectively. $Fe^{3+}$ peaks displayed two obvious peaks locating at BE of 712 eV and 725.3 eV, which were assigned to the Fe $2p_{3/2}$ and Fe $2p_{1/2}$ level [46]. In addition, 718.2 eV and

732.4 eV were ascribed to shake-up satellite peaks [26]. The above results demonstrated the existence of $Fe^{2+}$ and $Fe^{3+}$ in the $MnFe_2O_4$. The content of $Fe^{2+}$ was detected to be 32% and the content of $Fe^{3+}$ was 68%. For $MnFe_2O_4$ after use, the positions of the characteristic peak did not change. However, the total peak area decreased slightly after degradation, indicating that oxidation reaction occurred [28]. In accordance with the areas of two characteristic binding energy peaks, the contents of $Fe^{2+}$ was 34% and the content of $Fe^{3+}$ was 66% in the $MnFe_2O_4$ sample after activating PDS.

Similarly, in Figure 9c, the $MnFe_2O_4$ possessed Mn 2p peaks located at 641.5 eV and 653.2 eV, which correspond the Mn $2p_{3/2}$ and Mn $2p_{1/2}$ [48]. Figure 9c also displayed two distinct satellite peaks that were obtained from the Mn 2p spectrum, which was the signal of the $Mn^{2+}$ characteristic peak [18]. After degradation experiments, the Mn element in the catalyst stayed in the $Mn^{2+}$ state with no obvious changes.

In the end, peak fitting was performed for O 1s in $MnFe_2O_4$ samples before and after the reaction. Figure 9d showed the three peaks of O 1s spectrum of fresh samples were distributed at 529.7 eV, 531.8 eV and 533.4 eV, which were from surface lattice oxygen (Olatt), surface adsorbed oxygen (Oads) and adsorbed water (Owat), respectively [49]. In addition, three peaks were detected at 529.8 eV, 531.8 eV and 533.4 eV, respectively, after O 1s envelop decomposition of the used samples. The corresponding area ratio of adsorbed $H_2O$ increased from 16.47% to 25.98%. Compared with Olatt and Oads, the Owat possessed higher mobility.

In summary, this mechanism of the oxidative degradation of PCA by the $MnFe_2O_4$/PDS system was postulated, and the possible reaction processes are shown in Figure 10. First, the iron ions have higher catalytic activity to PDS in the octahedral sites of the $MnFe_2O_4$ spinel structure. Second, the active site undergoes a valence change at the solid–liquid interface, with iron and manganese ions as electron donors and $S_2O_8^{2-}$ as electron acceptors to produce $SO_4^{\bullet-}$, partially converting $S_2O_8^{2-}$ to $^1O_2$ and producing •OH. Finally, PCA reacts with various oxidation active substances [50].

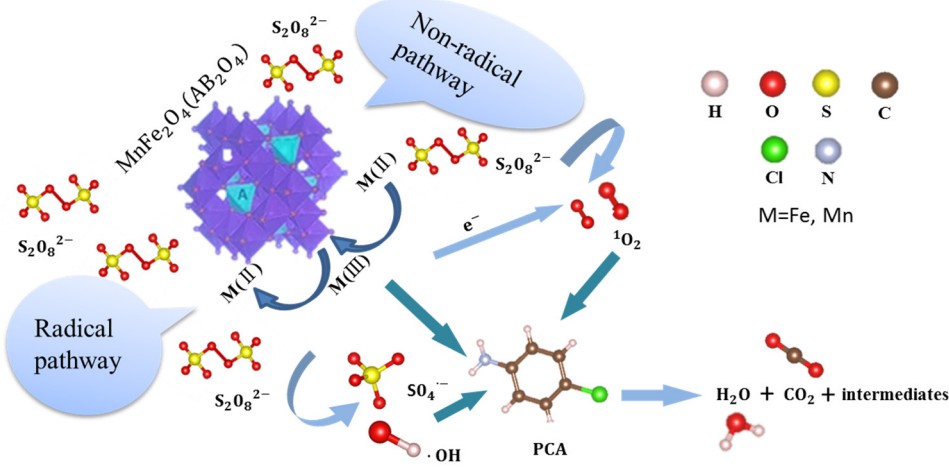

**Figure 10.** Schematic representation of the possible degradation mechanism of PCA in the $MnFe_2O_4$/PDS system.

## 4. Conclusions

In this work, $MnFe_2O_4$ was successfully synthesized by the chemical co-precipitation method. The catalyst was characterized by XPS, XRD, FTIR, TEM and SEM techniques. The efficiency of this catalyst in activating PDS for the degradation of PCA was investigated. The experiments showed that the decomposition of PCA by the $MnFe_2O_4$/PDS system depended on different reaction conditions (initial pH, reaction temperature, calcination temperature of catalyst). According to the quenching experiments, $SO_4^{\bullet-}$, •OH and $^1O_2$ issued from the $MnFe_2O_4$/PDS system were primary reactive oxygen species in the solution. Furthermore, the addition of more oxidant facilitated the mineralization of PCA. The PCA

degradation intermediates were identified with HPLC-MS, and based on the results, the possible transformation pathways for PCA degradation was proposed. It is concluded that PCA can be degraded by both free radicals and non-free radicals generated in the $MnFe_2O_4$/PDS system.

**Supplementary Materials:** The following supporting information can be downloaded at: https://www.mdpi.com/article/10.3390/pr10112227/s1, Table S1: Variation of solution pH in the process of PCA degradation by PDS catalyzed with $MnFe_2O_4$; Figure S1: SEM image (a) and TEM image (b) of $MnFe_2O_4$ material; Figure S2: The quasi-first-order dynamic model of PCA removal at different calcination temperature in $MnFe_2O_4$/PDS systems; Figure S3: The quasi-first-order dynamic model of PCA removal at different pH in the $MnFe_2O_4$/PDS systems; Figure S4: $pH_{pzc}$ of $MnFe_2O_4$, 500 °C; Figure S5: The quasi-first-order dynamic model of PCA removal at different temperature in the $MnFe_2O_4$/PDS systems. Figure S6: The quasi-first-order dynamic model of PCA removal by adding TBA in the $MnFe_2O_4$/PDS systems; Figure S7: The quasi-first-order dynamic model of PCA removal at by adding MeOH in the $MnFe_2O_4$/PDS systems; Figure S8: The quasi-first-order dynamic model of PCA removal by adding $NaN_3$ in the $MnFe_2O_4$/PDS systems.

**Author Contributions:** Conceptualization, B.L.; data curation, Y.S., P.M. and L.Q.; formal analysis, Y.S. and P.M.; funding acquisition, B.L.; methodology, L.Q.; supervision, B.L.; writing—original draft, Y.S. and L.Q.; writing—review and editing, Y.S., B.L. and P.M. All authors have read and agreed to the published version of the manuscript.

**Funding:** This research was funded by the Key Research and Promotion Project of Henan Province (No.222102320291).

**Institutional Review Board Statement:** Not applicable.

**Informed Consent Statement:** Not applicable.

**Data Availability Statement:** All data used to support the findings of this study are included within the article.

**Acknowledgments:** The authors would appreciate the funding support from the Key Research and Promotion Project of Henan Province (No.222102320291).

**Conflicts of Interest:** The authors declare that they have no conflict of interest regarding the publication of this paper.

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
