# Peer review of "Fe–Mn Oxide Composite Activated Peroxydisulfate Processes for Degradation of p-Chloroaniline: The Effectiveness and the Mechanism"

_processes, doi:10.3390/pr10112227_

Round 1

Reviewer 1 Report

This work describes the application of Fe-Mn oxides composite for p-chloroaniline (PCA) degradation under persulfate. Different perspectives on the impact on degradation of for p-chloroaniline, such as material systems, calcining temperatures, pH, temperature have been investigated. Additionally, the reaction mechanism of the Fe-Mn oxides composite for p-chloroaniline degradation with persulfate was investigated via trapping experiments, GC-MS measurement of intermediate. XPS was performed to characterized the catalytic materials before/after the catalysis. The quality of this work is poor. On both material and catalysis perspectives, I have major concerns about the scientific soundness and certain conclusion. I would recommend rejecting this work. 

1.     The authors should provide more characterization of Fe-Mn oxides composites, such as SEM, elemental mapping, XRD, etc. Now, there is basically no information of this Fe-Mn oxides composites, for example size. 

2.     The authors claimed that the Fe-Mn oxides composite was used as catalyst. What is the catalyst loading for the catalytic process? In the Figure, the concentration of PCA is 20 mg/L. 

However, the concentration of MnFe2O4 is 1.3 g/L. How come this is a catalytic process?

3.     What is the final product of p-chloroaniline degradation? Based on Figure 6, there are eight peaks on the LC spectra. Clearly, this is unacceptable and ridiculous being define as a catalytic degradation. 

4.     The quality of scientific language is also poor and the manuscript is missing necessary and enough reference, basically in every paragraph of introduction. 

5.     For example, in Figure 1, after 4 hours, the conversion is approaching to ~90 %. Why doesn’t the conversion go higher, in another word, to go completion?

6.     I am not convinced by any conclusion in the section of 3.5 Mechanism of PCA degradation by MnFe2O4/PDS systems. The authors performed GC-MS measurement to identify intermediate. Compound P1 and P2 were proposed based on GC-MS result. P1 and P2 are oxidated intermediates in the system, which unnecessarily proves that there are two mechanistic pathways. Furthermore, in figure 6, there are more than 8 species in this system and 6 of them are not identified. And I have no idea how the authors propose the steps such as ring cleavage, simple organic acid. Figure 8 and Figure 10 are not acceptable for a catalytic mechanism study.

Author Response

Dear Reviewer,

Thank you for your comments. Those comments are valuable and helpful for revising and improving our manuscript. We have studied comments carefully and made corresponding corrections which we hope meet with approval. Revised portion are marked in yellow in the manuscript.Please see the attachment.Please note:

  1. All changes (point by point) in the revised manuscript can be found

according to the following Response to the Reviewers' Comments.

  1. Black words are comments from Reviewers and red words are response to

the comments.

     3.Because of the modification of manuscript, the order of the references has

been updated in the revised manuscript.

We tried our best to improve the manuscript and made some changes in the manuscript. These changes will not influence the framework of the paper.

 We appreciate for Reviewers’ work earnestly, and hope that the correction will meet with approval.

Once again, thank you very much for your comments and suggestions.

                                                                                                      Best regards,

                                                                                                        Bingtao Liu

Reviewer 2 Report

Comments and suggestions:

1. Abstract- “…. reaction temperature and pH on the degradation of PCA were studied. In addition, the reaction mechanism of the MnFe2O4catalyzed oxidative degradation of PCA by PDSwas investigated by free radical quenching experiments, and the reactants leading to PCA degradation were identified using different scavengers. The results showed that singlet.” -The author should be clear in their writing whether it is removal or degradation of of p-chloroaniline.

2. “Therefore, direct discharge of wastewater containing PCA can cause environmental pollution, and it is crucial to find an efficient wastewater treatment method.”--- I suggest the author, to discuss a paragraph related to water pollution due to the presence of different contaminants and applications of different adsorbents for the treatment techniques. The authors are recommended to check the below related references, which will improve the supporting information.

Journal of Cleaner Production 241, 2019, 118263

Environmental research 2019, 170, 389-397

Journal of Molecular Liquids 317, 2020, 113916

Journal of environmental management 219, 2018, 285-293

Materials 5 (12), 2012 2874-2902

Desalination and Water Treatment 57 (46), 2016, 21863-21869

Nanomaterials 9 (5), 2019, 776

3. “The catalytic methods of PDSincludetransition metal activation[5], electrochemical activation [6], photo-activation [7], thermal activation [8] and microwave activation [9] ,etc..” – Delete etc.

4. “When the pH value is increased to 11, the efficiency of PCA was reduced by 43.88% compared with that of pH 7.”- Explain the possible reason. Separation Science and Technology 55 (10), 2020, 1766-1775

5. “To further investigate thedegradation mechanism of PCA in the MnFe2O4/PDSsystem, the intermediates during the oxidative degradation of PCA were identified by HPLC-MS.” – Various hyphenated HPLC methods should be defined in brief including their advantages.

International Journal of Environmental Analytical Chemistry 98 (2), 2018, 171-181

Arabian Journal of Chemistry 12 (5), 2019, 633-651

Journal of King Saud University-Science 32 (4), 2020, 2414-2418

6. “-xahydrate (FeCl3•6H2O) and manganese sulphate monohydrate (MnSO4•H2O)were supplied by Tianjin Comio Chemical Reagent Co.  Provide the details of the supplier and also do same for all chemical used including city and country in experimental section.

7. “For the preparation process and properties of the catalyst (MnFe2O4), please refer to the previous experimental results of their group ” – Change to “The preparation process and properties of the catalyst (MnFe2O4), were carried out based on the previous experimental results.”

8. “A liquid chromatograph (Waters W2489-QDa, USA) with a reversed-phase C-18 column (4.7 ×250 mm), the m”- Provide the details of the manufacturer including city and country and also do same for all instruments used.

9. “Figure6. Liquid chromatogram of PCA at 0 h and after 4 h of reaction”—Add the optimal conditions in each caption.

10. “Figure7.LC-MS identification of PCA degradation intermediates” – It was better to provide the separation peaks and mass spectrum in same figure to identify them in better way.

11. The English quality not up to the mark. All the typos and grammar need to check thoroughly in the manuscript, especially the uses of spaces must be corrected.

Author Response

Dear reviewer,

Thank you for your decision and constructive comments on my manuscript. We have carefully considered the suggestion of Reviewer and make some changes. We have tried our best to improve and made some changes in the manuscript.Please see the attachment.Please note:

  1. All changes (point by point) in the revised manuscript can be found

according to the following Response to the Reviewers' Comments.

  1. Black words are comments from Reviewers and red words are response to

the comments.

     3.Because of the modification of manuscript, the order of the references has

been updated in the revised manuscript.

We tried our best to improve the manuscript and made some changes in the manuscript. These changes will not influence the framework of the paper.

 We appreciate for Reviewers’ work earnestly, and hope that the correction will meet with approval.

Once again, thank you very much for your comments and suggestions.

                                                                                                      Best regards,

                                                                                                        Bingtao Liu

Reviewer 3 Report

The manuscript entitled “The removal efficiencies and mechanism of p-chloroaniline degradation by persulfate activated with Fe-Mn oxides composite”, by Yu Shi, Lin Qiao, Panfeng Ma, and Bingtao Liu,has been reviewed.

This manuscript investigates the removal of p-chloroaniline (PCA) using Fe and Mn oxide composite (MnFe2O4) catalysts prepared by chemical co-precipitation method. The MnFe2O4 catalyst and its activation on the decomposition of PCA was studied using X-ray Photo Electron Spectroscopy (XPS).The experiments showed that the decomposition of PCA by MnFe2O4/PDS system depended on several reaction conditions (initial pH, reaction temperature and the calcination temperature) of the catalyst. According to the quenching experiments, SO4•−, •OH and 1O2 dispensed from MnFe2O4/PDS system act as the primary reactive oxygen species for PCA degradation in solutions. Additionally, the TOC removal efficiency as a function of various amounts of oxidants is also investigated. The degradation intermediates of PCA were identified with HPLC-MS. It is concluded that MnFe2O4 is an efficient heterogeneous catalyst for the degradation of PCA by PDS.

The paper reports an efficient and advanced oxidation technology for waste water (water containing p-chloroaniline) treatment to eliminate or reduce the environmental pollution, which is definitely a burning issue of the century and hence worth publishable. The experimental analysis, such as synthesis, characterization and testing of the catalyst are very good. However, the presentation (writing) of the work especially the discussion part of the manuscript is very poor. The manuscript required a thorough revision, needs lot of corrections in English Grammar, sentence structure and flow of language.  Please do the corrections described below before publication.

Q1. P1, L09, “The results showed that singlet oxygen (1O2), sulfate radicals (SO4· – ) and hydroxyl radicals (•OH) may all be involved”.

Please correct the “SO4 ._” as (SO4•−) in the Abstract. Also please correct this on Page 2, L-21 as well. 

Q2. P2, L16-17, “The metal oxides are easily separated from the reacting liquid in subsequent processing.”

 Please correct the English grammar of the sentence.

Q3. P1, L22-23, “In recent years, spinel ferrites (e.g. CuFe2O4, CoFe2O4, MnFe2O4, etc.) have been widely concerned by researchers.”

Explain briefly why researchers are concerned about spinel ferrites?

Q4. P3, L7-P8, “For the preparation process and properties of the catalyst (MnFe2O4), please refer to the previous experimental results of their group [5, 16]”.

The authors mentioned that “the preparation process and properties of the catalyst (MnFe2O4)” are given in detail on references [5-16]. However for this context, please explain the synthesis process briefly (in two or three sentences).

Q5. P3, L-9-11, “20mg/L PCA was adjusted to the desired pH value with 0.1mol/L sulfuric acid or sodium hydroxide. Then a certain amount of catalyst and 50mL 20mg/L PCA were added to six 150mL conical bottles, respectively”.

Please correct the sentence structure

Q6. P3, L 24-25, “The concentration of PCA was analyzed by N-(1-naphthyl) ethylenediamine azo spectrophotometric method”

Please give a reference to this method.

Q7. P3, L29-34, “A liquid chromatograph (Waters W2489-QDa, USA) with a reversed-phase C-18 column (4.7 ×250 mm), the mobile phase of acetonetrile/ultrapure water (V/V= 55/45), the column temperature of 40 °C, the flow rate of 0.50 mL/min, the fragment ion scan range: 50-1050 amu, the mass spectrometer operated under positive ionization with electrospray at fragmenter voltage of 600V”

Check the spelling. It seems to be the author is listed all the testing parameters used/required to conduct the liquid chromatography experiment. Please write it into a meaningful sentence.

Q8. P4, L-3-6, “When MnFe2O4 and PDS exist in the reaction system, the removal efficiency of PCA gradually increased to 94% at 240 min. Further- more, the degradation efficiency of PDS/Fe3+ and PDS/Mn2+ systems for PCA in 240 minutes was 5.45% and 13.30%, respectively.”

Correct the sentence structure and English grammar.

Q9. P4, L-6-11, “The results showed that the oxidation mode of the MnFe2O4/PDS system is dominated by heterogeneous activation. And MnFe2O4 can be activated by PDS. Similarly, Jing et al [14] found that the removal of bisphenol A by homogeneous systems (PMS/Fe3+, PMS/Mn2+) waseless than 10%, which was much lower than the 90% removal by heterogeneous activation systems (MnFe2O4/PMS).”

Please correct the sentence structure and grammar.

How do you reach to a conclusion that the degradation of PCA was due to the oxidation of MnFe2O4/PDS system is dominated by heterogeneous activation? Is there any characterization results such as HPLC or FTIR that you can present to support your findings?

Q10. P4, L-15-16, “Relevant studies showed that the calcining temperature may affect the structure of the material itself, thus affecting its catalytic activity”.         

Please correct the “calcining temperature” as “calcination temperature”

Q11. P4, L-16-18, “In this work, the catalytic performance at different temperatures was compared, and the result was shown in Figure 2.”

Please correct the English grammar and the sentence structure.

Q12. P4, L-23-26, “These results showed that the removal rate decreases when the calcining temperature reached 600°C. This may be due to decomposition of the MnFe2O4 material to other oxidizing substances [17] (e.g. Fe2O3 and Mn2O3) at 600℃, which reduces the decomposition point and effective contact area for the oxidative degradation of PCA. It is not conducive to further effective removal of PCA.”

This part of the discussion is not clear. What is the decomposition temperature of PCA? Please elaborate your argument and correct the sentence structure.

Q13. P5, L10-11, “The maximum yield of PCA decomposition was achieved at pH 7, Kobs was 0.013 min-1 at pH 7”

Please define “Kobs

Q14. P5, L19-20, “The results might be cause by three reasons. First of all, with the initial pH increasing from 3.0 to 11.0, the leaching concentration of Fe3+ after 4h re- action were detected to be 0.12, 0.18, 0.14, 0.16 and 0.52 mg/L .”

Please correct the English grammar.

Q15. P6, L-9-11, “Finally, the charge of PCA and the surface charge of MnFe2O4 is also a crucial factor influencing the degradation of PCA in solutions”.

Please correct the English grammar.

Q16. P6, L-11-12, “It has been reported that the pKa value of PCA is 4.15”

Please define “pKa

Q17. Please correct the English grammar.

Please correct the English grammar.

Q18. P6, L14-16, “It indicated that the surface of catalyst had positive charges due to pH < pH pzc and negative charges in solutions with pH > pHpzc [22].

Please correct the English grammar.

Q19. P6, L39-41, “Where Ea was the activation energy (kJ mol-1), Kobs was the measured quasi-first-order dynamic constant, R was the universal gas constant (8.314J mol-1 k -1), A was Arrhenius constant and T was the temperature (K). According to Eq.5, the Ea of PCA degradation by MnFe2O4 was calculated to be 29.70 kJ mol-1. So the main thing in the MnFe2O4/PDS system was the chemical reaction”

Please correct the English grammar. 

In the present study, the activation energy, Ea of PCA degradation by MnFe2O4 is estimated as 29.70 kJ mol-1. Compare the result with a similar study and please give its reference.

Q20. P7, L4-6, “In order to study the degradation mechanism of PCA by MnFe2O4/PDS system, the trapping experiments was carried out to examine the main active species”.

Please correct the English grammar. 

Q21. P7, L4-6, “In order to study the degradation mechanism of PCA by MnFe2O4/PDS system, the trapping experiments was carried out to examine the main active species.”

Explain briefly about “the trapping experiments”. Give reference.

Q22. P7, L7-9, “MeOH has high heat of reaction for both SO4 •− and •OH (kSO4 •−= 3.2×106 M -1 s -1, k•OH = 9.7×108 M -1 s -1). Moreover, TBA can react to •OH with more fast kinetic rates”

Please define “MeOH” and “TBA”

Q23. P7, L19-20, “Besides the main radicals of SO4 •− and •OH, non-radical like 1O2 may simultaneously be produced in PDS involved oxidation system (Eq.7-8) [25]”.

Please correct the English grammar. 

Q24. P7, L21-22, “As shown in Figure 4 (c), NaN3 is used to quench 1O2 [26]. 1O2 played the significance role in PCA oxidation (from 92.16% to 60.80%)”.

Please correct the English grammar. 

Q25. P7, L27, Side heading, “3.4. TOC removal efficiencies in MnFe2O4/PDS system”

Please elaborate “TOC” in the heading and give TOC in parentheses (TOC).

Q26. P7, L37-38, “The possibility of further formation of inorganic or organic small molecule oxidation products and retention in solution.”

Please complete the sentence meaningfully.

Q27. P7, L37-38, “Structural features of compounds determine the solubility, molecular arrangement, spatial structure, chemical functional grous, intermolecular attraction and repulsion of organic compounds. This affects whether the organic matter can be catalyzed and the degree of degradation.”

Please correct the sentence structure and correct the spelling.

Q28. The authors fail to present the experimental procedure of the following experiments in the Materials and Methods section.

Give a brief experimental procedure of

3.3  Quenching experiments of PCA degradation

3.4 TOC removal by chemical treatment method

In two or three sentences in the “Materials and Methods section”.

Author Response

Dear Reviewer,

Thank you for your comments. Those comments are valuable and helpful for revising and improving our manuscript. We have studied comments carefully and made corresponding corrections which we hope meet with approval.We apologize for the poor language of our manuscript. We have now worked on both language and readability. We really hope that the flow and language level have been substantially improved. Revised portion are marked in yellow in the manuscript.Please see the attachment.Please note:

  1. All changes (point by point) in the revised manuscript can be found

according to the following Response to the Reviewers' Comments.

  1. Black words are comments from Reviewers and red words are response to

the comments.

     3.Because of the modification of manuscript, the order of the references has

been updated in the revised manuscript.

We tried our best to improve the manuscript and made some changes in the manuscript. These changes will not influence the framework of the paper.

 We appreciate for Reviewers’ work earnestly, and hope that the correction will meet with approval.

Once again, thank you very much for your comments and suggestions.

                                                                                                      Best regards,

                                                                                                        Bingtao Liu

Reviewer 4 Report

The manuscript described persulfate activation using MnFe2O4. Metal oxide based catalysts for persulfate activation was already explored and this manuscript does not provide any new insight in this field. However, the results are interesting and further modification of the manuscript is required. 

The specific comments are as follows:

(i) Why MnFe2O4 was chosen? What is the role of Mn2+? The author should compare the activity of MnFe2O4 with Fe3O4 to justify the replacement of tetrahedral sites of a spinel.

(ii) It is not clear why composite oxides are formed at a certain temperature.

(iii) The potentials for different radicals should be clearly mentioned in the manuscript.

(iv)   Controlled experiments with pure Mn2O3 and Fe2O3 are required. 

(v) The mechanism of reaction has been provided without any clear evidence.

(vi) Recent references on disulfate activation should be cited:

 https://doi.org/10.1002/adsu.202000288, J. Mater. Chem. A 20208, 15513.Appl. Catal., B 2020261, 118214.

Author Response

(The authors gave the same response as above.)

Round 2

Reviewer 1 Report

I would like to thank the authors making efforts to address my comments. I recommend the editors rejecting this paper based on two of my comments:

In comment #2, the catalytic loading of this processing was questioned. As a catalytic paper, the catalytic loading should always be identified and provided.

Secondly, I have major concern about the mechanism and two pathways that the authors provided. 

a.     Figure 10 is too poor to be a scheme for a demonstration of catalytic mechanism. Please refer to a real catalysis paper how to provide catalytic mechanism;

b.     The two pathways that the authors have provided are not conclusive at all and are based on the GC-MS results in the first draft, which, however, was removed in the second draft. In the GC-MS spectrum in the first draft, there are more than 8 species in the reaction and many are not identified. 

To me, this paper is not good enough to be published in Process. The authors should submit to other low-impact journal. 

Author Response

Dear reviewer,

Thank you again for your decision and constructive comments on my manuscript. We have carefully considered the suggestion of Reviewer and make some changes. We have tried our best to improve and made some changes in the manuscript.

Please note:

  1. All changes (point by point) in the revised manuscript can be found according to the following Response to the Reviewers' Comments. Please see the attachment.
  2. Black words are comments from Reviewer and red words are response to the comments.
  3. Because of the modification of manuscript, the order of the references has been updated in the revised manuscript.

We tried our best to improve the manuscript and made some changes in the manuscript. These changes will not influence the framework of the paper.

We appreciate for Reviewers’ work earnestly, and hope that the correction will meet with approval.

Once again, thank you very much for your comments and suggestions.

Best regards,

Bingtao Liu

Reviewer 4 Report

As most of the comments are addressed y the authors, the manuscript can be accepted.

Author Response

Dear reviewer,

Thank you for your approval of our modifications. We apologize for the poor language of our manuscript. We now make one more correction to the grammar. We really hope that the language level has been substantially improved. For the references, we have also updated. The abstract and conclusions of the manuscript have been newly described, allowing for a clearer display of research results.

We appreciate for Reviewers’ work earnestly. Once again, thank you very much for taking time out of your busy schedule to read and revise our manuscript.

Best regards,

Bingtao Liu

Round 3

Reviewer 1 Report

I am pleased with the author responses to specific questions, they have clarified all of those.